# A Review of the Biomimetic Structural Design of Sandwich Composite Materials

**DOI:** 10.3390/polym16202925

**Published:** 2024-10-18

**Authors:** Shanlong Che, Guangliang Qu, Guochen Wang, Yunyan Hao, Jiao Sun, Jin Ding

**Affiliations:** 1Naval Architecture and Port Engineering College, Shandong Jiaotong University, Weihai 264209, Chinazszh6157@163.com (G.Q.);; 2School of Mechanical, Electrical and Information Engineering, Shandong University, Weihai 264209, China

**Keywords:** additive manufacturing, biomimetic structure design, interlayer interface optimization

## Abstract

Sandwich composites are widely used in engineering due to their excellent mechanical properties. Accordingly, the problem of interface bonding between their panels and core layers has always been a hot research topic. The emergence of biomimetic technology has enabled the integration of the structure and function of biological materials from living organisms or nature into the design of sandwich composites, greatly improving the interface bonding and overall performance of heterogeneous materials. In this paper, we review the most commonly used biomimetic structures and the fusion design of multi-biomimetic structures in the engineering field. They are analyzed with respect to their mechanical properties, and several biomimetic structures derived from abstraction in plants and animals are highlighted. Their structural advantages are further discussed specifically. Regarding the optimization of different interface combinations of multilayer composites, this paper explores the optimization of simulations and the contributions of molecular dynamics, machine learning, and other techniques used for optimization. Additionally, the latest molding methods for sandwich composites based on biomimetic structural design are introduced, and the materials applicable to different processes, as well as their advantages and disadvantages, are briefly analyzed. Our research results can help improve the mechanical properties of sandwich composites and promote the application of biomimetic structures in engineering.

## 1. Introduction

Sandwich composite materials comprise face sheets and a core layer. These materials play crucial roles in aerospace, marine engineering, and cushioning packaging due to their light weight and high strength [1]. A suitable core layer structure design can effectively reduce the product damage inflicted by impacts and vibrations [2,3]. Over billions of years of evolution in their respective environments, natural organisms have gradually developed structures and functional properties that are highly adapted to their surroundings, exhibiting excellent mechanical properties [4]. Biomimetic structures use natural designs to simplify mechanical properties and enhance performance. This innovative approach, integrating various biomimetic structures, represents a new direction in achieving diverse mechanical enhancements. In recent years, biomimetic structural design has provided a new and effective method for improving the comprehensive mechanical properties of composite materials [5,6]. Examples include biomimetic thermal insulation teacups inspired by the calla lily [7], biomimetic actuators based on the mechanism of the Venus flytrap [8], biomimetic solutions to the integration of soft and hard materials modeled from shells [9], and biomimetic surfaces emulating lotus leaves for surface pollution removal [10]. Drawing from natural organisms, researchers currently focus on core layer structures from two main perspectives: an optimized design based on a single biomimetic structure and an integrated design based on multiple biomimetic structures. With the goal of achieving synergies between materials and biomimetic structures, the diversity of composite materials can be combined to obtain different properties through the use of materials like acrylonitrile butadiene styrene (ABS), polylactic acid (PLA), and poly ether-ether-ketone (PEEK) as matrices and carbon fiber as reinforcement.

Improving the interfacial bonding strength between the face and core layers of different materials is one of the key issues affecting the mechanical properties of sandwich composite materials. Currently, heterogeneous materials can be combined through two main methods: adhesive bonding and interlocking structure connection. Adhesive bonding is the simplest, safest, and most cost-effective method. It features uniform stress distribution, a light weight, minimal surface corrosion, sound insulation, and low vibration. In the development of adhesives, Chen et al. [11] mirrored the structure of the human skeletal muscle. They created a high-performance biomass adhesive with significant practical application potential by using various green reinforcing materials. Chen et al. [12] patterned their work after the adaptive hydrophobic and hydrophilic interactions of mussel foot proteins to improve soy protein adhesives, significantly enhancing their bonding strength. However, adhesive bonding still has the disadvantage of experiencing a reduced bond strength over time and in different working environments. Interlocking structures utilize their interlocking nature to achieve a connection among heterogeneous materials instead of adhesive bonding. Accordingly, the potential for a reduced bond strength due to time or environment is avoided. Xie et al. [13] drew inspiration from plant cell structures and utilized ceramic and graphite as the cytoplasm and cell wall layers to design and prepare biomimetic honeycomb-structured ceramic/graphite composite materials. Li et al. [14] studied the exoskeleton of beetles and achieved a good combination of strength and ductility between soft and hard materials. The present article analyzes the impact of biomimetic structures on the mechanical properties of sandwich composite materials from the perspective of biomimetic structural design. We summarize the research in three aspects, namely, biomimetic structural design, the interfacial bonding between the face and core layers, and the manufacturing of biomimetic sandwich composite materials. Finally, the current challenges and development prospects of sandwich composite materials based on a biomimetic structural design are discussed.

## 2. Core Layer Biomimetic Structural Design

Over billions of years of evolution in their respective environments, natural organisms have gradually developed structures and functional properties that are highly adapted to their surroundings. Thus, they exhibit excellent mechanical properties [4]. Given the limited resources of organisms, the structures they have evolved are the ones with the highest degree of adaptability [15]. Therefore, biomimetic structures can provide a new and effective method of designing core layer structures in sandwich composite materials [5].

### 2.1. Animal Biomimetic Structures

As a commonly used biomimetic structure, the honeycomb structure has tremendous potential in energy absorption and impact resistance. By filling different fibers and designing anisotropic honeycomb structures, the overall stiffness of the honeycomb structure is greatly improved [16,17]. Traditional honeycomb structures such as square, hexagonal, and circular exhibit significantly different mechanical properties under different wall thickness conditions [18]. Inspired by biomimetic materials, Xu et al. [19] proposed a variable-thickness honeycomb with enhanced biomimetic cells. Many other biomimetic studies on animal structures have been performed. For example, Yang et al. [20] prepared a body-centered cubic lattice structure and studied the compressive properties of functionally graded squid-like lattice materials. Li et al. [21] proposed a new type of biomimetic multicellular tube that simulates the unique double diagonal reinforcement structure in the glass sponge cell.

A sandwich composite material often uses different materials for the face and core layers, and the shell-like biomimetic structure perfectly combines hard and soft materials. Shell nacre is a natural material with high strength and toughness. This excellent performance is primarily related to its “brick-and-mortar” microstructure comprising inorganic substances and a small volume fraction of organic substances. The cohesion and sliding of aragonite tablets and the bridging effect of organic substances are the key factors influencing their high toughness and crack arrest [22,23]. Through in-depth research on the shell structure, a 3D model of its basic structure is created, as shown in Figure 1. On the one hand, this simple structure can minimize its own weight. On the other hand, it ensures longitudinal strength without reducing the axial strength through staggered stacking.

The two-way corrugated structure inspired by the mantis shrimp is also a promising direction for resisting impact and energy absorption. By utilizing the excellent impact resistance of the mantis shrimp’s two-way corrugated structure, the properties of the core material can be improved. To this end, the key structure of the mantis shrimp is extracted, and the corrugated structure is taken as the research object for further study. Han et al. [25] proposed an aluminum honeycomb–corrugated hybrid core structure, showing advantages in compressive strength and energy absorption, especially under low-density conditions. Zhang et al. [26] studied the dynamic behavior of different types of mantis shrimp-inspired corrugated sandwich structures. They found that the structure with a flat core had the smallest maximum contact force and maximum back deflection. Hou et al. [27] conducted experimental and numerical simulations on multilayer corrugated sandwich panels under quasi-static crushing loads. They concluded that the yield stage is prolonged with increased corrugated sandwich layers, leading to better energy absorption before densification.

For sandwich composite materials, the two-way corrugated structure is favored in core material design due to its excellent impact resistance. Yang et al. [28] established a biomimetic model (Figure 2) by observing the surface structure of mantis shrimp fingertips. They used it as a core material to test its impact performance. The bidirectional ripple structure can dissipate more energy and has better energy absorption abilities. Compared with the traditional unidirectional corrugated structure and triangular structure, the specific energy absorption of the bidirectional corrugated structure increases by 60% and 58.6%, respectively. These features can better realize structure stability and also reduce material consumption.

### 2.2. Plant Biomimetic Structures

In addition to studying the biomimetic structures of animals, the unique structures of plants are also worthy of analysis. For instance, the pearl-layer structure [29], the bamboo structure [30,31], horsetail plants [32], and biomimetic carambola structures [33] all exhibit diverse mechanical properties. Beyond the research on the single performance of biomimetic structures, Xu et al. [34] conducted biomimetic research on the lattice structure of lotus roots. They demonstrated an outstanding performance in flexibility and hardness.

Pomelo peel is known to act as a natural protective barrier for its internal flesh and seeds due to its porous layered structure that dissipates energy, ensuring the protection of the fruit’s interior when it falls [35,36,37]. Through experimental research on pomelo peels, Fischer et al. [38] proved that pomelo peels possess good impact resistance. By analyzing the microstructure of pomelo peels, a model is established, as shown in Figure 3.

For comparison with traditional honeycomb structures, a finite element model was built, as depicted in Figure 4. Compared with traditional honeycomb structures, the pomelo peel honeycomb structure can absorb more energy when subjected to axial forces, thereby allowing it to withstand greater pressure. Experimental analysis [39] revealed that the pomelo peel honeycomb structure can improve energy absorption by approximately 30% compared with traditional honeycomb structures when subjected to axial forces. The reason is that the layered honeycomb has an earlier densification stage, which can realize the concentrated combination of an in-plane and out-plane crushing force. Meanwhile, for a traditional honeycomb and teak honeycomb with the same density, the outside crushing force of the teak honeycomb is 1.5 times higher than that of the traditional honeycomb, and the in-plane crushing force is even 2.5 times higher. This finding indicates that structural changes can significantly enhance the compression resistance and the energy absorption performance. In the engineering field, the good energy absorption properties of the teak skin biomimetic structure are effective in cushioning and shock proofing, as well as in guiding subsequent engineering applications of the multilevel material.

Wood cells are a typical spiral composite structure comprising rigid spiral cellulose microfibrils and a soft matrix of hemicellulose and lignin [40], as shown in Figure 5. The spiral structures rotating at different angles within them further enhance the impact resistance of the entire structure. In a study on wood cells, Fratzl et al. [41] established a simplified mechanical performance model for softwood cells. Deng et al. [42] utilized molecular dynamics (MD) simulation techniques to build a digital model of wood cell walls. By designing different porous structures, spiral angles, and spiral directions, they explored the constitutive relationship between the structure of wood and its mechanical properties.

Experiments have shown that a smaller spiral angle in the structure corresponds with a higher stiffness and elastic modulus. Conversely, a larger spiral angle corresponds with a stronger torsional strength and deformation ability. Owing to this characteristic, the change in the helix angle is utilized to enhance the mechanical properties while maintaining the same quality, which can be well adapted to high-strength tubes and support columns and other application scenarios. This finding can also explain why different parts of wood throughout its life cycle can change the angle of microstructural fibrils and thus achieve good mechanical properties based on its needs.

### 2.3. Fusion of Multi-Biomimetic Structures

The fusion of multi-biomimetic structures primarily utilizes the characteristics of different biomimetic structures and integrates them to enhance material properties. Spiderweb structures dissipate up to 70% of dynamic impact energy in the air [43,44], whereas the S-shaped structure of cuttlefish bones exhibits highly efficient energy absorption in the deep sea [45,46,47]. The porous structure of pomelo peels possesses excellent cushioning properties, effectively protecting the pomelo fruit [48]. Inspired by the S-shaped structure of cuttlefish bones, spiderweb structures, and the porous material in pomelo peels, integrating these three structural materials into a biomimetic design can further enhance the energy absorption and cushioning performance of materials [49]. As shown in Figure 6, experiments have revealed that this hybrid structure significantly improves the energy absorption and shock reduction performance of the material.

During finite element analysis (FEA), stress concentration points inevitably occur in the simulation of a single structure, regardless of the specific structure. However, the integration of multiple biomimetic structures can significantly mitigate this issue, leading to a more stable structure, as illustrated in Figure 7.

In addition to leveraging the cushioning properties of pomelo peel, the spiderweb structure also contributes to improving the energy absorption efficiency. Sethi et al. [50] studied the impact resistance of a naturally inspired bamboo–spiderweb hybrid structure based on theoretical and numerical simulation methods. Their results showed that the proposed novel fourth- and third-order hybrid cellular lattice elements could effectively resist impact loads. Mousanezhad et al. [51] designed a novel spiderweb-like layered honeycomb structure and investigated its in-plane elastic response under large and small deformations through theoretical, simulation, and experimental methods. He et al. [52] designed a novel honeycomb structure based on the spiderweb and studied its out-of-plane compression properties through experimental and numerical simulation. This novel structure is found to be more effective in enhancing energy absorption than traditional honeycomb structures.

Additionally, Ji et al. [53] combined spiderwebs with cellular structures to design an angle-enhanced spiderweb honeycomb, as shown in Figure 8. This spiderweb can evenly distribute stress, utilizing radial lines and ribs at the corners to distribute the load uniformly over the surface. Accordingly, equal loads are ensured on both sides of the cell walls and the central nodes. Consequently, the joint loading of each cell wall is achieved, resulting in excellent mechanical properties.

Biomimetic structures can reduce the use of materials and ensure stable performance. The comparative analysis in Table 1 shows that the integration of multiple biomimetic structures has obvious advantages over traditional biomimetic structures. In terms of energy consumption, the angular reinforced spiderweb honeycomb structure can better disperse the stress, and the fracture energy absorption is as high as 1494 kN·mm. In terms of the tensile strength, the biomimetic structure inspired by the cell wall of wood is able to improve its tensile performance due to its internal helical structure, and the highest tensile strength can reach 34.9 MPa. The bidirectional corrugated structure can evenly distribute stress to each contact point during compression, so the compression strength reaches 690 MPa, which is 13 times higher than that of the biomimetic structure of the wood cell wall. However, due to the different materials used, some deviation may exist between the experimental data and the actual performance. Therefore, the specific performance data need to be determined according to the materials used. By comparing the different biomimetic structures mentioned above and describing their respective advantages and disadvantages, representative biomimetic structures of animals, plants, and fusion biomimetics are summarized for a more intuitive understanding of the differences between different biomimetic structures, as shown in Figure 9.

## 3. Surface–Core Interface Bonding

The interface bonding between the core layer structure and the surface panel directly affects the overall mechanical properties of the sandwich composite material. In the current research, the bonding between complex biomimetic sandwich core structures and surface panels is primarily achieved through two methods, namely, adhesion and interlocking structures.

### 3.1. Adhesion

Adhesion primarily utilizes adhesives to achieve bonding between different structures and layers. The adhesive achieves good interfacial bonding between the substrate and reinforcing materials through wetting and bonding effects. García-Guzmán et al. [62] studied long-fiber composite materials. They found that the mechanical properties of trapezoidal pattern interfacial adhesive joints have a resilient and promising performance. Sun et al. [63] analyzed the impact of pattern geometry on the fracture behavior of adhesive joints. Huang et al. [59] studied the use of catechin structures that mimic the high adhesion of mussel proteins in soybean protein-based adhesives to improve adhesion effects. Chen et al. [64] utilized environmentally friendly biomass polyurethane derived from epoxidized soybean oil as an adhesive. They found an improvement in adhesive strength, thereby achieving environmental protection goals. Liu et al. [65] used 3-aminopropyltriethoxysilane to achieve adhesion in a sandwich structure. They reacted the -NH_2_ groups on the activated wood surface with the epoxy groups in the adhesive. The adhesive strength was significantly enhanced after the boiling water treatment. The adhesion method was simple to operate, but different adhesives needed to be used for different materials. Adhesives play a crucial role in composite material manufacturing and have a significant impact on the overall mechanical properties of composite materials.

Adhesives can more effectively transmit stress between the components of composite materials. When composite materials are subjected to external forces, adhesives can help transfer stress from one material to another, achieving overall stress distribution and enhancing the load-bearing capacity of composite materials. Adhesives can also affect the damage tolerance of composite materials. When composite materials are subjected to impact or damage, adhesives can absorb and disperse energy to a certain extent. The outcome is a reduced impact on the overall performance. Adhesives can further enhance the toughness of composite materials, making them less prone to brittle fracture when subjected to external forces. However, the selection and use of adhesives also affect the durability, weatherability, chemical resistance, and environmental performance of the composite material. The careful selection and use of adhesives is required when preparing sandwich composites. To ensure the optimum performance of the composite material, a necessary consideration when selecting an adhesive is its chemical nature. The adhesive must not react with the composite material and generate new substances that lead to a poor bonding effect.

### 3.2. Interlocking Structure

An interlocking structure is a design that utilizes specific connections to lock different layers of composite materials together. This structure provides additional stability and strength, enabling multilayer composite materials to exhibit a superior overall performance when subjected to external loads or environmental factors. Compared with adhesion, interlocking structures have significant advantages in terms of their interlayer bonding force and bonding effect. Lin et al. [66] studied the mechanical properties of different biomimetic composite material interlocking stitch interfaces. Theoretical and experimental results have shown that triangular stitch interfaces, due to their uniform stress distribution, can produce high levels of stiffness, strength, toughness, and integrity. Inverse trapezoidal geometries also exhibit mechanical interlocking with damage tolerance and higher toughness. Li et al. [67,68] compared the mechanical properties of ordinary stitch lines and hierarchical stitch lines, finding that increasing the number of layers within the triangular stitch lines can significantly improve the mechanical properties. Fractal serrated interlocking designs also reportedly enhance tensile strength [69] and fracture toughness [70,71]. Interlocking interfaces with multiple stable structural configurations are bound to exhibit excellent strength and toughness [72,73].

The armored beetle, also known as the hard-shelled beetle, possesses an extremely sturdy shell that safeguards its delicate internal organs. Research on the shell of the armored beetle has led to the discovery of a similar interlocking interface in its multifunctional exoskeleton. This interlocking structure can support loads that are 39,000 times greater than the beetle’s body weight [74]. By applying biomimetic modeling to its exoskeleton, an interlocking structure resembling a trapezoid was obtained, similar to the one depicted in Figure 10.

In the study of interlocking connection structures, Rivera et al. [74] researched four sets of joints with an identical panel thickness, core thickness, and core density. To control variables, a hexagonal honeycomb structure made of the same material was utilized as the primary structure of the core material. A traditional solid core, adhesive-bonded core, Type A joint core, and Type B joint core were designed separately and subjected to displacement shear experiments. The experiments revealed that the use of interlocking structures is superior to adhesive bonding in terms of the cost and connection performance.

Lin et al. [66] conducted a biomimetic study on the suture interfaces of diatom attachment zones. They summarized four representative interlocking geometries, namely, inverse trapezoid, rectangle, trapezoid, and triangle. To ensure the accuracy of the comparative experiments, mechanical performance tests were performed under the condition that only the shape varied while the other factors remained the same. This approach enabled a better analysis of the optimal interface bonding shape.

After comparing the experimental data, the triangular geometric shape was found to be the optimal choice due to its ability to uniformly distribute stress, resulting in high stiffness, strength, ductility, and integrity. The inverse trapezoidal suture interface is also desirable because it exhibits high stiffness, strength, and ductility when equipped with an initially bonded tip interface, providing a natural advantage in interface bonding.

Compared with adhesive bonding, interlocking structures exhibit superior mechanical properties. Interlocking structures achieve tight connections between layers through physical means rather than by relying solely on the function of adhesives, as in adhesive bonding. This physical connection provides a more stable and reliable structure, reducing the risk of connection failure due to adhesive aging, failure, or environmental factors. Furthermore, interlocking structures allow layers to work synergistically when subjected to external loads, thereby enhancing the overall mechanical properties of the composite material. For instance, under tensile or compressive loads, interlocking structures ensure the even distribution of loads between layers, preventing stress concentration and localized damage. Interlocking structures can also improve the impact resistance and fatigue resistance of composite materials, enabling them to maintain a stable performance in various complex working environments.

### 3.3. Interface Bonding Optimization

Apart from experimental methods, using MD principles to analyze intermolecular forces and investigate their effect on the composite surface is also one of the methods used to optimize the improvement. Song et al. [76] fabricated Al_2_O_3_/graphite laminates by embedding Al_2_O_3_ layers into graphite, enhancing the bonding strength between layers and boosting the in-plane bending strength by 10 times compared with graphite blocks. Xie et al. [13] designed a honeycomb biomimetics-based mesocarbon microbead (MCMB) particle and tungsten carbide (WC) composite material, thereby significantly improving its mechanical properties. As illustrated in Figure 11, variations in the MCMB-WC content alter particle interactions and the stress-induced performance disparities, leading to distinct fracture patterns. Farnoosh et al. [77] tailored surface properties by adjusting the polyvinylpyrrolidone/vinyl acetate and PLA content and droplet size. Mihaela et al. [78] explored the synergy between wheat gluten and xanthan gum to boost composite performance and understand its binding mechanisms. For bio-ink materials, fibroin protein stands out due to its exceptional mechanical properties and biocompatibility. Sakai et al. [79] delved into the activity and mechanical behavior of fibroin nanofibers during printing processes. Adding fibroin protein does not reduce the cell activity or affect the cell behavior, and the mechanical properties are also improved with an increased fiber density. The feasibility of replacing cellulose nanofibers is also discussed.

The MD simulation method is also one of the ways to study the intermolecular force. Han et al. [80] used the MD simulation method to study the interface adhesion properties of an aggregate and asphalt binder, deepening our understanding of the interface-bonding mechanism and verifying the model’s accuracy. Chen et al. [81] used a MD analysis method to determine the effective curvature of molten metal nanodroplets and realized the exploration of surface tension at the nanoscale. John et al. [82] used machine learning to simulate MD, calculate, and train interatomic energy and forces. They revealed the atomic mechanism involved in producing graphene on diamond surfaces.

Moreover, inverse analysis provides a powerful tool for this purpose. Inverse analysis involves inferring unknown parameters or properties from observational data or experimental results. Incorporating artificial intelligence (AI) and nature-inspired techniques into inverse analysis can significantly improve its accuracy and efficiency. Chen et al. [83] mention the use of the approximate Bayesian calculation (ABC) method to establish a mapping relationship between various parameters and their corresponding properties. By sampling from the posterior distribution, ABC can approximate the true distribution of parameters given the observed data, thereby enabling more reliable inverse analysis. M.E. et al. [84] used mechanical control and design optimization techniques to design constitutive equations and governing equations that use geometric parameters and the volume fraction of particle phases as design variables. This approach enables the optimization of shell performance by fine tuning these parameters to achieve the desired mechanical properties. Zhang et al. [85] demonstrated the effectiveness of coupling FEA with machine learning techniques to predict the buckling strength of composite materials with high accuracy. On the one hand, FEA can obtain a large number of data sets; on the other, machine learning can realize fast prediction. The combination of the two shows high sensitivity, increasing the results’ accuracy. Currently, FEA is commonly used for interface optimization simulations. Utilizing ANSYS, a model is established to represent the key characteristics and behaviors/functions of the sandwich composite’s interlocking structure. It is systematized and formulated for simulation and analysis. Taking the inverse trapezoidal interlocking structure as an example, the simulation results obtained through FEA are illustrated in Figure 12. After collecting and comparing the simulation results with the experimental data, certain discrepancies are found between them. These discrepancies may be due to the environmental factors and material stability that require further improvement. However, comparing the mechanical properties of the two different interlocking structures with adhesive bonding shows the feasibility of replacing the traditional adhesive process with an interlocking structure.

Machine learning can also be used to optimize the interface bonding of composite structures. It can learn from a large amount of experimental data and predict the mechanical properties of multilayer composite materials. By training machine learning models, complex relationships among material compositions, process parameters, and properties can be established, enabling the rapid screening and optimization of new materials. Elizabeth et al. [87] utilized machine learning to predict and analyze the shear strength and flexural strength of materials, optimizing performance through their excellent fitting degree and achieving the equivalent substitution of simulations. Cai et al. [88] used artificial neural networks to analyze the interlayer and out-of-layer bonding strength of continuous ramie fiber-reinforced polypropylene composite materials. The interface strength was found to be positively correlated with the mechanical properties of the composite materials. A higher extrusion flow rate, lower layer thickness, and higher printing speed improve the interface strength. Meanwhile, through a horizontal comparison between the three prediction algorithms, the artificial neural network algorithm has better prediction accuracy. Yuan et al. [89] successfully predicted the interfacial shear strength and interfacial fracture toughness of poly (p-phenylene benzobisoxazole) fiber-reinforced epoxy resin composite materials using kernel-ridge regression. Machine learning can also assist in identifying the optimal structural configuration during the design process of multilayer composite materials. By training machine learning models to learn under different structural configurations, the rapid search and evaluation of biomimetic structural designs can be achieved. Ultimately, the optimal design that meets performance requirements can be discovered.

Inverse analysis, combined with AI and nature-inspired techniques, provides a powerful framework for evaluating mechanical properties. The ABC method, mechanical control and design optimization, and the coupling of FEA with machine learning can be leveraged to improve the accuracy and efficiency of mechanical property evaluations. As these methods continue to evolve, they are likely to play an increasingly important role in advancing engineering and scientific research on the mechanical optimization of biomimetic structures.

## 4. Manufacturing of Sandwich Composite Materials Based on Biomimetic Structural Design

The processing and preparation methods for biomimetic sandwich composite materials vary depending on their material composition. Traditional methods require the separate preparation of the core layer and the face layers, involving multiple steps and a long production cycle. However, with the continuous development of additive manufacturing technology, the integrated formation of sandwich composite materials has been achieved. It promotes the widespread application of biomimetic sandwich composite materials in various fields, such as engineering and medicine [90,91,92].

### 4.1. Selective Laser Sintering (SLS)

SLS is one of the most commonly used additive manufacturing technologies due to its ability to print parts with a good surface finish, dimensional accuracy, mechanical properties, and high durability [93]. It allows for the production of parts without the use of molds, making it ideal for prototyping and small-batch production [94]. Additionally, SLS does not require any support structures, eliminating the need for post-processing after the printed part is removed. After sintering all layers, the sintered pieces are extracted, excess powder is removed, and the remaining powder can be recycled for new work [95]. SLS technology has fewer limitations in the selection of powders, allowing for the sintering and printing of metallic and non-metallic materials. Nevertheless, the materials utilized in SLS technology are predominantly metal powders, limiting its applicability to those with inadequate thermal stability or sluggish cooling rates. At the same time, because of powder printing, the surface of SLS molding is often rough and porous, the bonding strength between the powders is limited, and pores and cracks easily form, affecting the mechanical properties of the bionic structure. Thus, additional post-processing operations are required, further increasing the manufacturing cost and time. Additionally, during printing, the polymer material in the sintering volatilizes odor gas. Consequently, the presence of dust can also lead to deviations in the bionic structure during the printing process and uneven printing results.

Regarding the impact of SLS technology on printing performance, Li et al. [33] designed a biomimetic structure based on the five-pointed star shape of the cross-section of a carambola fruit, which exhibits excellent impact performance. Using nylon as the main material, the specimens were printed by SLS technology, and non-axial compression experiments were performed to determine its good energy-absorbing performance and deformation stability. Verma et al. [96] tested the compressive strength of a novel biomimetic lattice structure under SLS. They also tested the flow rate inside the structure and its overall compressive properties by using PA12 material as a simulation material and further investigated the powder recycling in SLS printing. Moreover, Yan et al. [97] utilized SLS to prepare carbon-fiber-reinforced PEEK composite materials to study their tensile strength, processability, and reusability, thereby providing ideas for the diversification of materials used in SLS technology.

### 4.2. Stereolithography (SLA)

SLA is one of the earliest rapid prototyping technologies, emerging in the 1980s. During its subsequent development, various extensions of SLA digital light processing have emerged. Vafaeefar et al. [98] used SLA technology to print three different lattice biomimetic structures combined with a highly flexible resin to study their mechanical behavior, energy absorption, and deformation patterns under high compressive deformation. Tavangarian et al. [99] developed a biomimetic model with natural needle-like structures and used rigid resin as the material for rapid printing by using SLA technology to analyze and fabricate brittle rods with nested cylindrical structures featuring high strength and toughness. Chen et al. [100] used SLA printing to characterize the three-dimensional structure of the beetle forewing using a pure trabecular beetle sheath plate model and using a honeycomb plate model as a comparative model. Its mechanical properties and failure mechanism were investigated after printing with ABS resin, and it was found that the compressive strength and energy-absorbing capacity of the beetle sheath plate model was two to three times higher than that of the honeycomb plate. Chen et al. [101] used SLA for 3D printing to conduct a biomimetic study of bone structures, as well as to improve the compressive strength and stability of the sponge structure by controlling the pore size. SLA printing avoids the generation of dust, enables high-precision printing, and has a smooth printing surface, which is a great advantage for printing fine parts and enables the printing of complex bionic structures. However, due to the limitations of the printing method, most of the selected materials are photosensitive resins, resulting in greater material limitations and higher costs. They may also not be as good as metal materials or other composite materials in terms of strength, stiffness and other properties, and when testing for high-strength bionic structures, it may be the case that the materials are unable to measure the ultimate force on the structure. Meanwhile, due to the special characteristics of photosensitive materials, SLA printing requires a high working environment, rendering its large-scale application in real life impossible and increasing the maintenance cost. When printing, designing a support structure is usually necessary to maintain stability and integrity. The subsequent removal of the support structure is very likely to cause damage to the molded part, which could very easily lead to the destruction of the bionic structure and ultimately invalidate the experimental data.

### 4.3. Fused Deposition Modeling (FDM)

FDM involves melting long filaments and depositing them layer by layer through an extruder nozzle onto a print bed. The molten filaments quickly cool, solidify, and adhere onto the printer. The printing process continues for the next layer until the entire object design is completed. FDM enables the rapid prototyping of complex biomimetic structures, and the development of multi-material FDM provides new solutions for the integrated formation of sandwich composite materials with faces and cores [102]. Concurrently, owing to the printing limitations of FDM technology, the majority of metal materials are unfeasible, rendering polymer materials the most suitable and optimal solution.

Compared with other rapid prototyping technologies, FDM technology can better achieve the integrated formation of biomimetic sandwich composite materials. Zhu et al. [103] printed a model with a volcano biomimetic structure by using PLA as a raw material and using FDM technology, through which the rate of seawater desalination increases. Ganesh et al. [104] conducted a biomimetic study on the arrangement of fish scales, using FDM technology and thermoplastic polyurethane as a raw material. They fabricated an aircraft shell imitating the arrangement of fish scales to reduce air resistance. Tao et al. [105] performed the preparation of willow biomimetic structures by mixing wood particles and PLA as a raw material through FDM. They found that the biomimetic wood structures had good compression resistance. FDM technology has unique advantages in printing complex structures, but it also has some unavoidable limitations. They include its low mechanical strength, high porosity, interlayer adhesion defects, and strong anisotropy in the printed parts [106]. FDM printing, from equipment to materials, has a lower price, utilizes materials more efficiently to reduce material waste, and improves efficiency. When choosing materials and designs that have a higher degree of freedom, FDM printing has a unique advantage due to its bionic structure, the printing process does not produce harmful substances, it produces little pollution, a low threshold is used, and the process can be used in large-scale applications. However, due to the limitations of the printing method itself, the printing accuracy and surface quality are limited, which may result in the experimental results of the bionic structure not meeting expectations; in addition, the printing speed is slow, which affects the production efficiency. Removing the support structure after the completion of the printing is also difficult, which could very easily lead to the destruction of the bionic structure and ultimately invalidate the experimental data; in addition, the limited performance of the material used for printing restricts its application in specific environments.

For current additive manufacturing technology, the types of materials that can be used in additive manufacturing are still too few. Different manufacturing technologies also have more stringent requirements regarding the types of materials used, and some materials can only be employed in one additive manufacturing technology. Additionally, with the further development of digital twins and machine learning, the self-correction of models and real-time defect detection during printing are areas of future research.

## 5. Conclusions

Sandwich composite materials based on biomimetic structures apply natural biological structures to practical engineering practices, achieving the complementary advantages of different structures and materials while ensuring excellent mechanical properties. Biomimetic structures also have excellent performance in energy consumption and pressure resistance compared with solid materials, and biomimetic structure sandwich materials consume lower energy while reducing the mass and improving the overall quality of the product.

Biomimetic structures can ensure the mechanical properties of materials due to their light weight, and the biomimetic design of multistructure fusion can also achieve the outstanding optimization of certain mechanical properties. The interlocking structure has a better performance in the connection of the interface of heterogeneous materials, and the connection between materials is realized through different biomimetic structures. This greatly improves the overall mechanical properties of composite materials and also realizes the goal of green environmental protection and sustainable development. Determining how to achieve the optimization and improvement of composite materials, compared with experiments, simulation or machine learning, plays an important role in optimizing and improving the mechanical properties of composite materials. Given the continuous development of 3D-printing technology, new methods and materials continue to emerge, providing more possibilities for printing. Composite and multi-material printing between two different materials is also a future trend, and the complementary properties of multi-materials can improve the overall performance. Further research is needed on the effect of process parameters on the overall mechanical properties of sandwich composites.

## Figures and Tables

**Figure 1 polymers-16-02925-f001:**
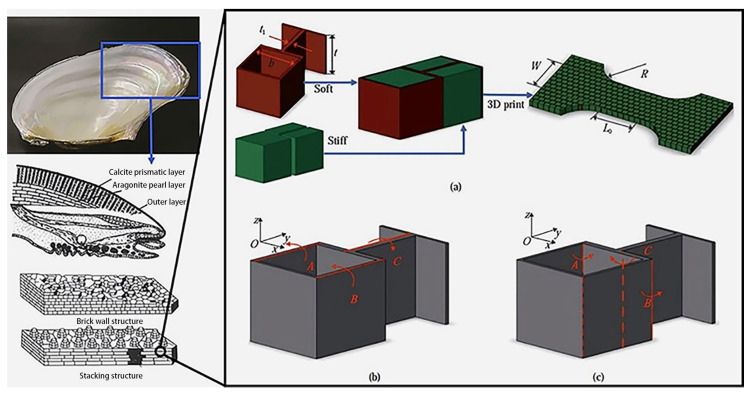
Biomimetic structural design of shell nacre (**a**) imitation shell pearl layer structure preparation (**b**) soft material and specimen face in the direction of the angle (**c**) soft material and specimen face outside the direction of the angle [24].

**Figure 2 polymers-16-02925-f002:**
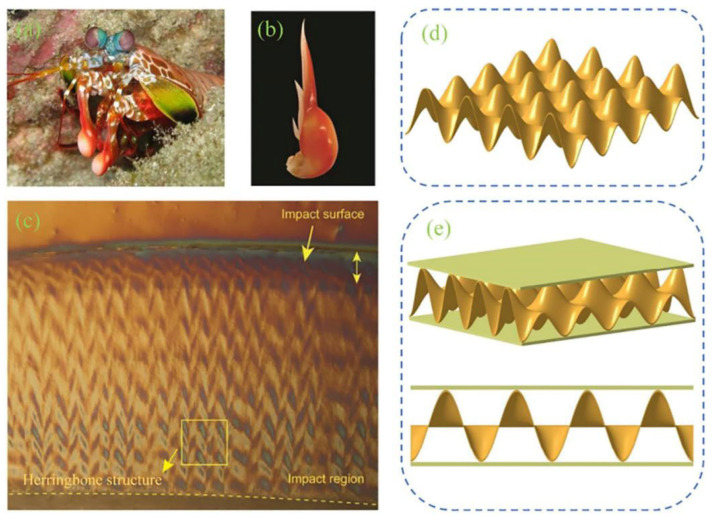
Biomimetic design of the surface structure and model of the mantis shrimp: (**a**) representative appearance of Odontodactylus scyllarus with two integrated dactyls; (**b**) individual drawing of the dactyl club; (**c**) CT scanning of a dactyl club section; (**d**) bio-inspired bi-directionally sinusoidal corrugated panel; (**e**) bio-inspired bi directionally sinusoidal corrugated sandwich structure [28].

**Figure 3 polymers-16-02925-f003:**
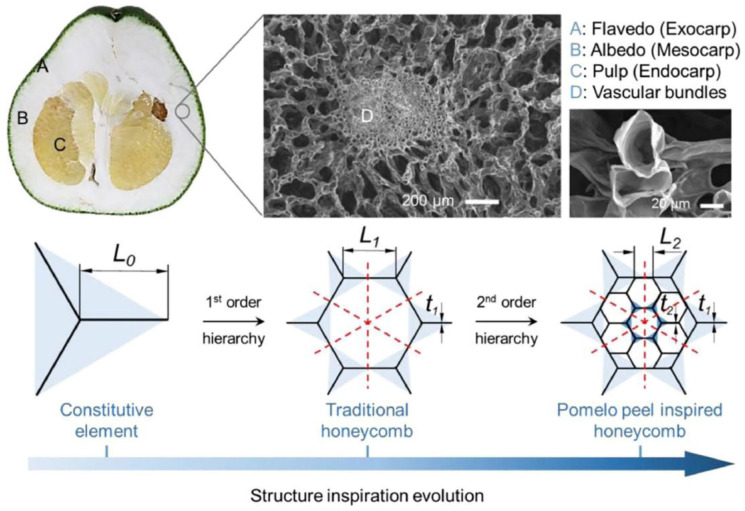
Detailed microstructure and geometric model of pomelo peel [38].

**Figure 4 polymers-16-02925-f004:**
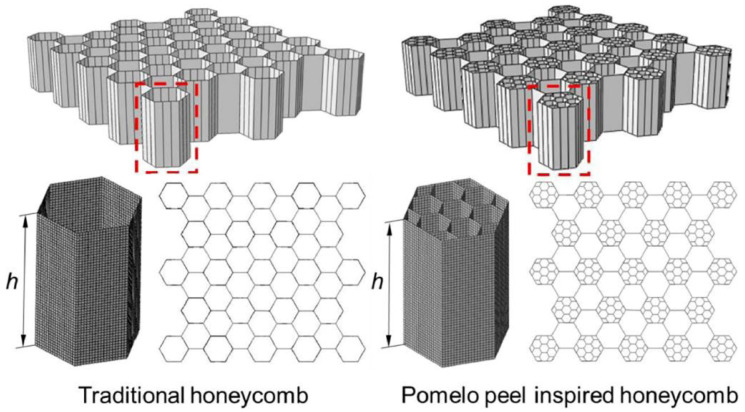
Finite element models of traditional honeycomb and pomelo peel honeycomb [38].

**Figure 5 polymers-16-02925-f005:**
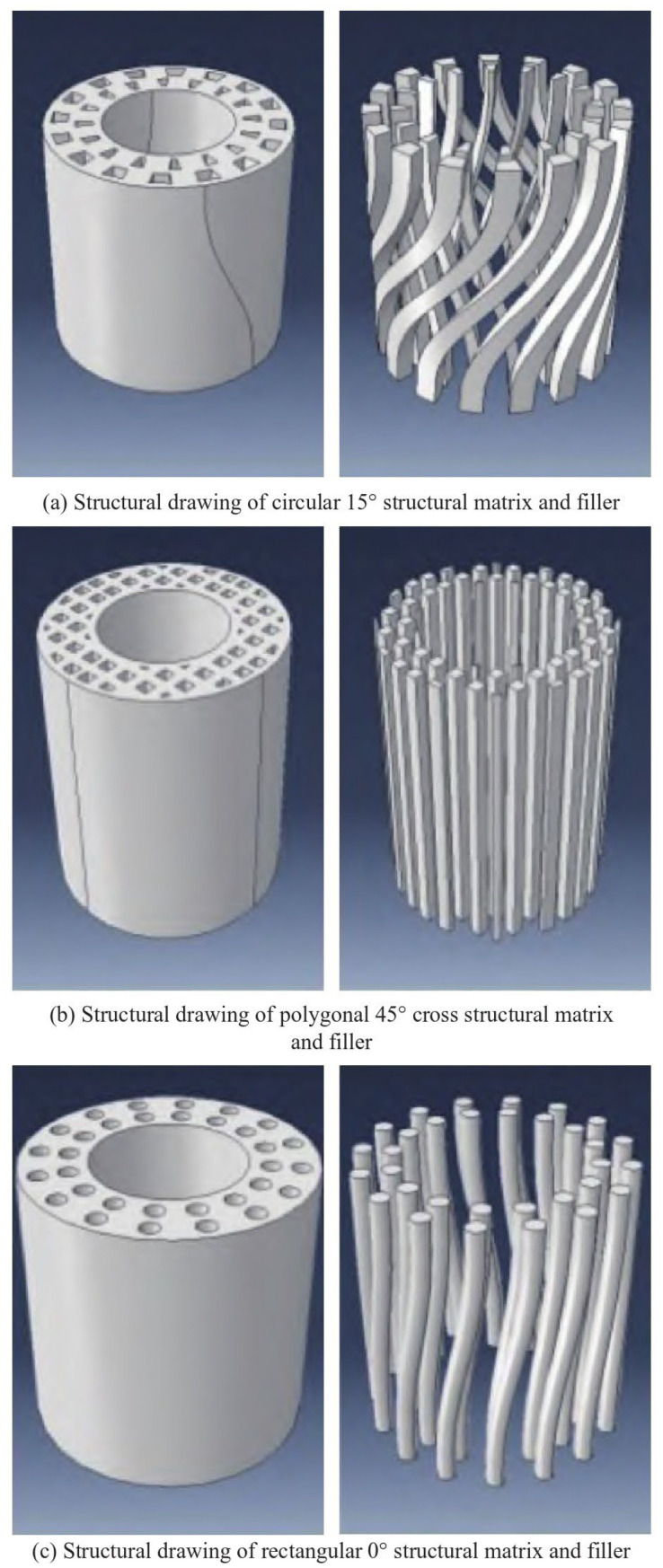
Biomimetic structural model of wood cell walls with different spiral angles [42].

**Figure 6 polymers-16-02925-f006:**
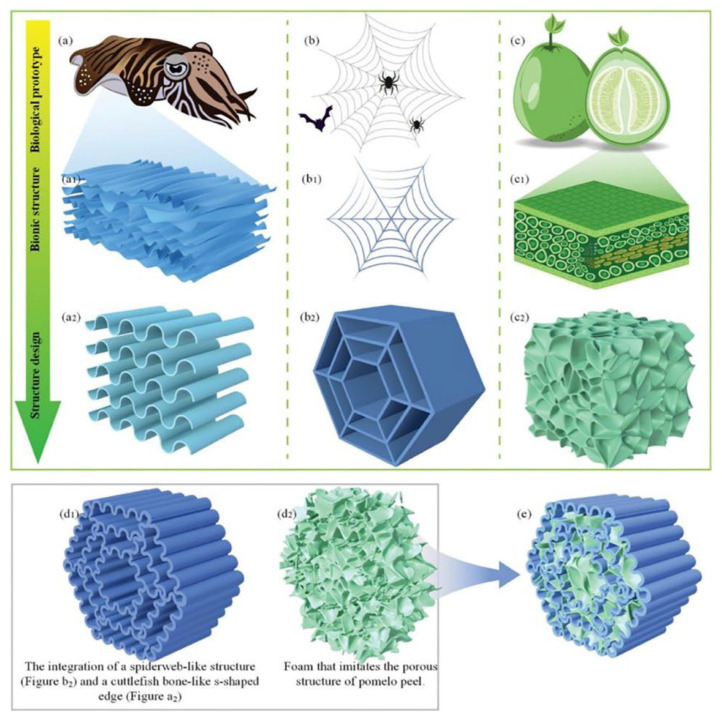
Biological prototypes and design strategies for the fusion of biomimetic designs. (**a**,**a1**) Schematic diagram of the S-shaped structure of cuttlefish bone. (**a2**) Bionic design of the S-shaped structure of cuttlefish bone. (**b**) Spider web, the second biological prototype used for integrated bionic design. (**b1**) Schematic diagram of the structure of spider web. (**b2**) Bionic structure of spider web. (**c**) Pomelo, the third biological prototype used for integrated bionic design. (**c1**) Schematic diagram of the porous structure of pomelo peel. (**c2**) Bionic design of the porous structure of pomelo peel. (**d1**) The integration of a spiderweb-like structure (**b2**) and a cuttlefish bone-like s-shaped edge (**a2**). (**d2**) Foam that imitates the porous structure of pomelo peel. (**e**) The S-shaped spider web integrated with porous foam to form a flexible composite of porous materials with an S-shaped spider web structure that was designed in this study [49].

**Figure 7 polymers-16-02925-f007:**
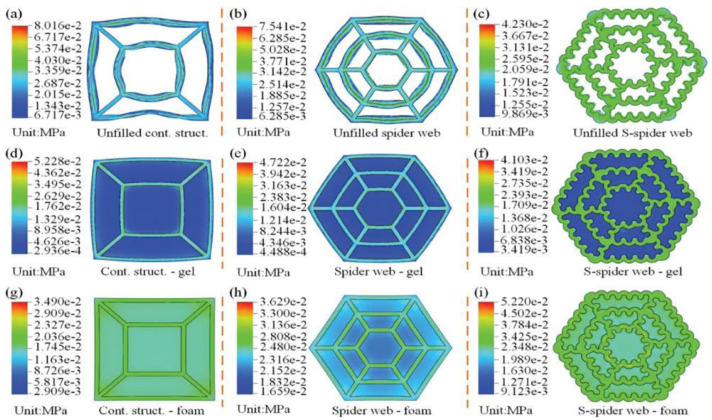
Finite element analysis results of impact mechanical properties of the integrated bio-inspired structure (**a**) FE simulation results of an unfilled contrast structure. (**b**) FE simulation results of an unfilled spider web. (**c**) FE simulation results of an unfilled S-spider web. (**d**) FE simulation results of a contrast structure-gel. (**e**) FE simulation results of a spider web-gel. (**f**) FE simulation results of an S-spider web-gel. (**g**) FE simulation results of a contrast structure-foam. (**h**) FE simulation results ofa spider web-foam. (**i**) FE simulation results of an S-spider web-foam [49].

**Figure 8 polymers-16-02925-f008:**
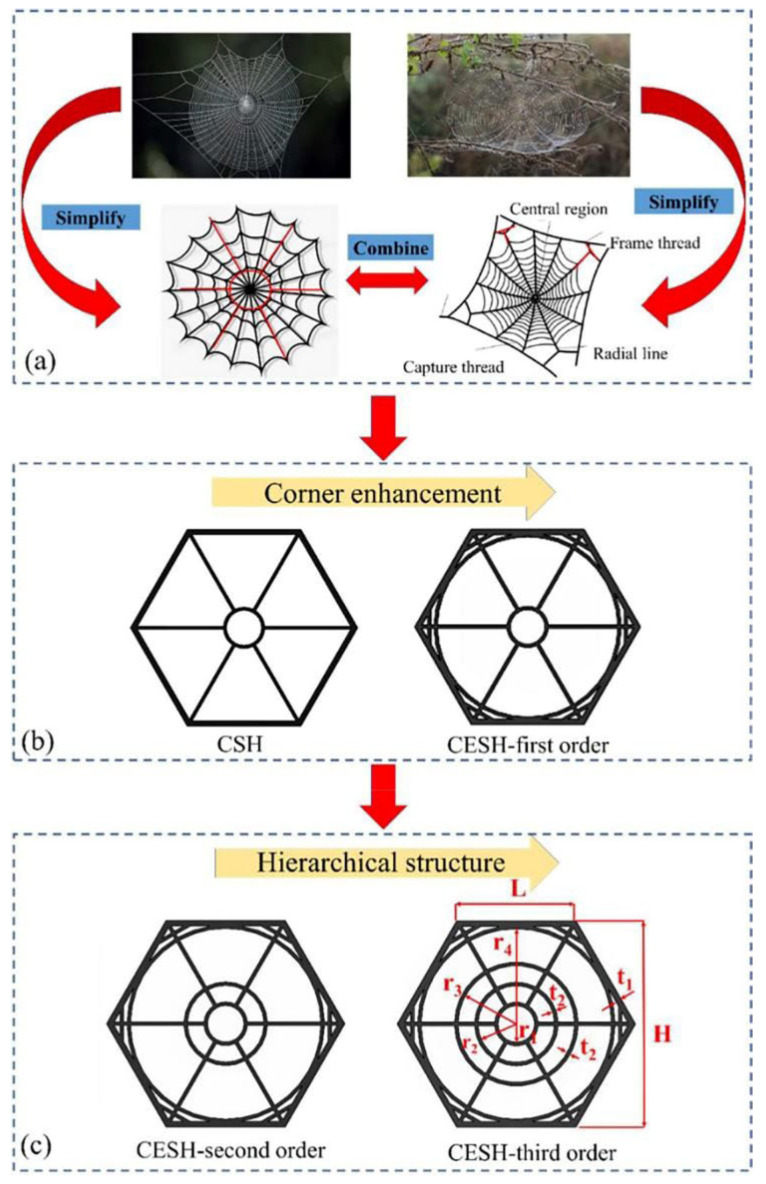
Design of angle-reinforced spider web honeycomb biomimetic structure. (**a**) Natural spiderweb structure and its simplified representation; (**b**) corner- enhanced biomimetic spiderweb-structured honeycomb models; and (**c**) corner-enhanced biomimetic spiderweb hierarchies [53].

**Figure 9 polymers-16-02925-f009:**
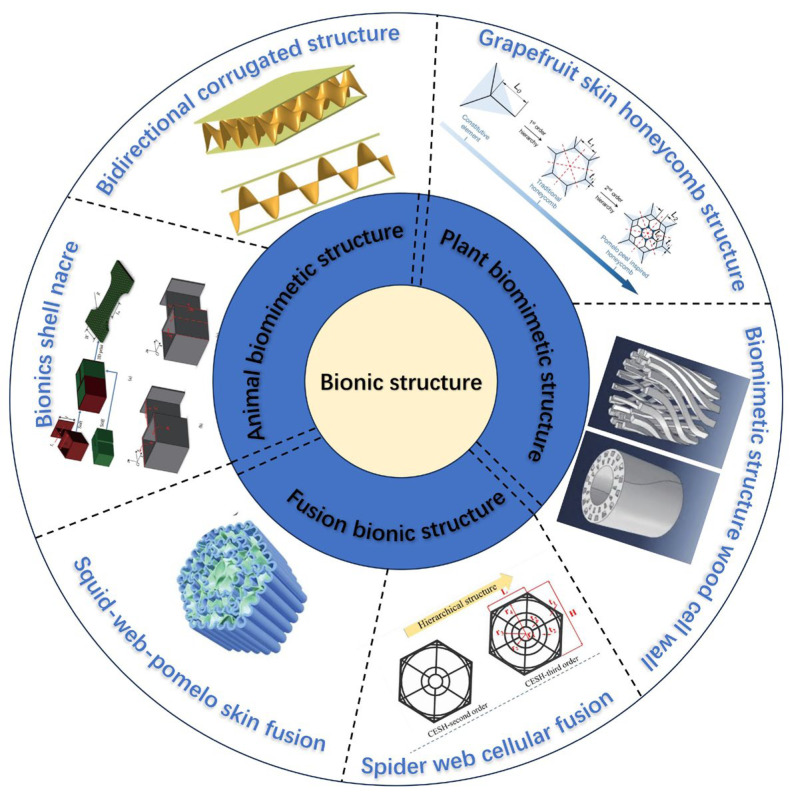
A review of the typical design of different biomimetic structures.

**Figure 10 polymers-16-02925-f010:**
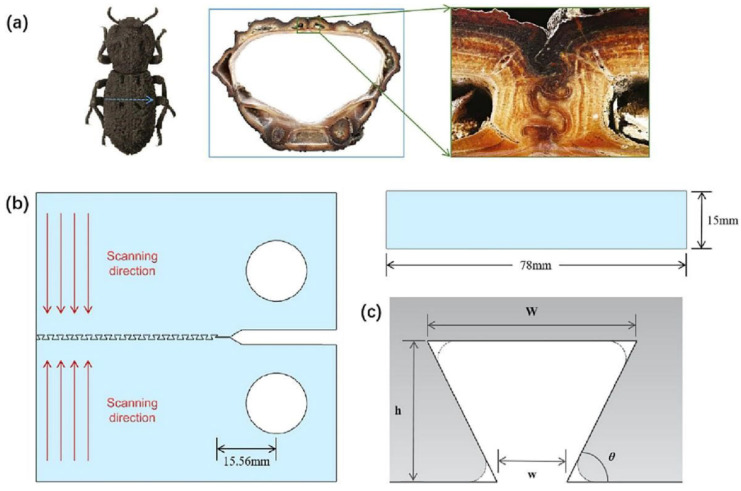
Design of interlocked structure imitating the exoskeleton of the armored beetle (**a**) biological suture interface; (**b**) design of the specimen model; (**c**) parameters of inter locked tooth [75].

**Figure 11 polymers-16-02925-f011:**
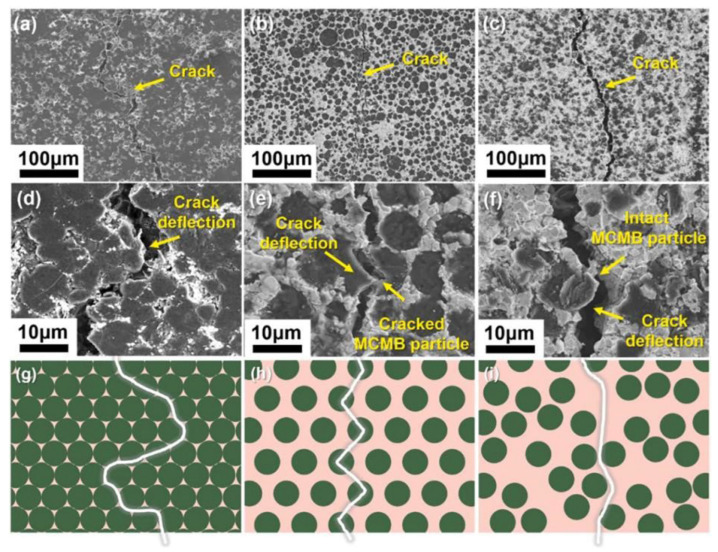
(**a**–**c**) Low-power SEM images of MCMB@WC crack growth path under different volume fractions: (**a**) 17%; (**b**) 53%; (**c**) non-biomimetic structure; (**d**–**f**) high-power SEM images of MCMB@WC crack growth path under different volume fractions: (**d**) 17%; (**e**) 53%; (**f**) non-biomimetic structure; (**g**–**i**) schematic diagram of MCMB@WC crack growth under different volume fractions: (**g**) 17%; (**h**) 53% and (**i**) non-biomimetic structures [13].

**Figure 12 polymers-16-02925-f012:**
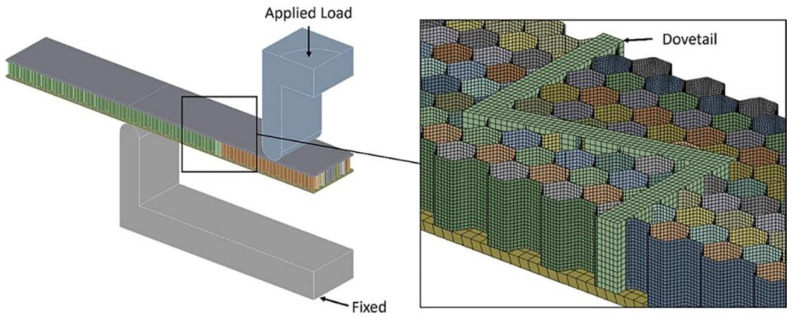
Finite element analysis of the bending performance of interlocked structures [86].

**Table 1 polymers-16-02925-t001:** Impact of biomimetic structures on mechanical properties.

	Fracture Energy Absorption (N·mm)	Tensile Strength (MPa)	Compressive Strength (MPa)	Ultimate Force (N)	Fracture Toughness (MPa·m^1/2^)	Flexural Strength (MPa)	Fracture Work (KJ/m^2^)
Shell Nacre Biomimetic Structure	423 [28]	22.5 [24]	—	37.5 [54]	1.9 [55]	4.8 [55]	0.9 [55]
Mantis Shrimp Bidirectional Corrugated Structure	287.2 [28]	74 [28]	690 [56]	45,000 [28]	—	—	199.8 [57]
Pomelo Peel Honeycomb Structure	2300 [58]	—	0.131 [58]	7500 [58]	—	—	25.6 [58]
Wood Cell Wall Biomimetic Structure	74,000 [59]	34.9 [38]	52.7 [38]	3000 [60]	—	80 [61]	—
Squid–Pomelo Peel–Spider Web Integrated Biomimetic Structure	4800 [49]	1.36 [49]	0.211 [49]	600 [49]	—	—	—
Angle-Reinforced Spider Web Honeycomb Structure	1,494,000 [53]	11.8 [53]	0.64 [53]	2900 [53]	—	13.7 [53]	—

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
