# Peer review of "A Review of the Biomimetic Structural Design of Sandwich Composite Materials"

_polymers, 2024, doi:10.3390/polym16202925_

Round 1

Reviewer 1 Report

Comments and Suggestions for Authors

1.       The definition of bioinspired regarding the developed in the study materials should be better explained to the readers.

2.       The most of the reviewed methods provide specific material patterns and possibilities for printing. Provide some limitation of the methods regarding reproducing mimicking structures available in nature.

3.       It is reported for example that ECM structure is reproduced by electrospinning, e.g. reported elsewhere doi.org/10.3390/polym14030529, how the authors have chosen the structure to mimic, what was the logical pathway and scientific novelty of the expected results.

4.       There are always some deviations of the structures obtained experimentally with the theoretical ones. Especially in case of additive manufacturing is used, these differences can be quite significant. In addition, post-processing to remove the residual powder particles exists, as in case of SLS.

5.       In case of sandwich structures, the molecular bonding between the adjacent layers should be discussed.

Reviewer 2 Report

Comments and Suggestions for Authors

Report on polymers-3258352

Dear editor

I am pleased to review the MS entitled: "A Review of the Mechanical Properties of Sandwich Composite Materials Based on Biomimetic Structural Design".

The Review discussed the Sandwich composite materials are extensively used in the engineering field due to their excellent mechanical properties. Accordingly, the interface bonding problem between their panels and core layers has always been a hot research topic. The emergence of biomimetic technology has enabled the integration of the structure and function of biological materials from living organisms or nature into the design of sandwich composites, greatly improving the interface bonding and overall performance of heterogeneous materials.

It seems interested and presents overview on the Biomimetic Structural Design trace but needs some modifications before considering for publication as follows:

-Title worth to improve to: A Review of the Biomimetic Structural Design of Sandwich Composite Materials Based.

-Abstract to short, it must be put in more details to clear the topics discussed in the review.

-Section 2. Core-Layer Bionic Structural Design needs more details and more works discussed it.

-The pomelo peel as natural protective barrier and wood cell as spiral composite structure comprising rigid spiral cellulose microfibrils and need some applications and what is their relation with Bionic Structural Design?.

-Conclusion must contains a summarized about all sections discussed and a comparison between them and their relations by the review topics, also, if possible the practical results will contribute in structures design and industries.

-Refs. 116, too large, the authors try to reduce it by removing the related refs. 

-Table 1 must be cited or more discussion needed.

-Figures put in increasing resolution.

Comments on the Quality of English Language

English is not fit and needs revising by native professor in english.
